# CoVigator—A Knowledge Base for Navigating SARS-CoV-2 Genomic Variants

**DOI:** 10.3390/v15061391

**Published:** 2023-06-17

**Authors:** Thomas Bukur, Pablo Riesgo-Ferreiro, Patrick Sorn, Ranganath Gudimella, Johannes Hausmann, Thomas Rösler, Martin Löwer, Barbara Schrörs, Ugur Sahin

**Affiliations:** 1TRON—Translational Oncology at the Medical Center of the Johannes Gutenberg-University Mainz Gemeinnützige GmbH, 55131 Mainz, Germany; 2BioNTech SE, 55131 Mainz, Germany; 3Research Center for Immunotherapy (FZI), University Medical Center of the Johannes Gutenberg University Mainz, 55099 Mainz, Germany

**Keywords:** SARS-CoV-2, dashboard, genomic variants, software, pipeline, virus genome assemblies, knowledge base, intrahost

## Abstract

Background: The outbreak of the severe acute respiratory syndrome coronavirus-2 (SARS-CoV-2) resulted in the global COVID-19 pandemic. The urgency for an effective SARS-CoV-2 vaccine has led to the development of the first series of vaccines at unprecedented speed. The discovery of SARS-CoV-2 spike-glycoprotein mutants, however, and consequentially the potential to escape vaccine-induced protection and increased infectivity, demonstrates the persisting importance of monitoring SARS-CoV-2 mutations to enable early detection and tracking of genomic variants of concern. Results: We developed the CoVigator tool with three components: (1) a knowledge base that collects new SARS-CoV-2 genomic data, processes it and stores its results; (2) a comprehensive variant calling pipeline; (3) an interactive dashboard highlighting the most relevant findings. The knowledge base routinely downloads and processes virus genome assemblies or raw sequencing data from the COVID-19 Data Portal (C19DP) and the European Nucleotide Archive (ENA), respectively. The results of variant calling are visualized through the dashboard in the form of tables and customizable graphs, making it a versatile tool for tracking SARS-CoV-2 variants. We put a special emphasis on the identification of intrahost mutations and make available to the community what is, to the best of our knowledge, the largest dataset on SARS-CoV-2 intrahost mutations. In the spirit of open data, all CoVigator results are available for download. The CoVigator dashboard is accessible via covigator.tron-mainz.de. Conclusions: With increasing demand worldwide in genome surveillance for tracking the spread of SARS-CoV-2, CoVigator will be a valuable resource of an up-to-date list of mutations, which can be incorporated into global efforts.

## 1. Introduction

The identification, characterization and monitoring of the pathogen responsible for a novel emerging disease is crucial for the development of a timely public health response. This includes rapid and open sharing of data [1], which has been adapted in past outbreaks to advance research and improve medical support [2,3]. The outbreak of the respiratory disease COVID-19 caused by SARS-CoV-2 demonstrated the increasing value of high-throughput sequencing through enabling the publication of the complete virus genome within one month of sampling [4,5]. The identification of the SARS-CoV-2 spike protein as a valuable target for vaccine design [6,7] led to the development of vaccines at unprecedented speed [8,9] and is still fostering further developments.

Nevertheless, the discovery of SARS-CoV-2 spike-glycoprotein mutants, associated with the potential to escape vaccine-induced protection, demonstrates the importance of monitoring SARS-CoV-2 genomic sequences to enable early detection of these genomic variants of concern. In a first study, we analyzed 1,036,030 genomic assemblies from the Global Initiative on Sharing Avian Influenza Data (GISAID) [10,11,12] and 30,806 Next Generation Sequencing (NGS) datasets from the European Nucleotide Archive (ENA). We reported non-synonymous spike protein mutations and their frequencies and analyzed the effect on known T-cell epitopes [13]. Although we confirmed low mutation rates of the spike protein, we experienced an increase in the number of genomic variants over time. Therefore, we further developed our computational pipeline to tackle potential escape mutations [14].

There are multiple initiatives to monitor SARS-CoV-2 mutations based on the GISAID dataset: NextStrain [15], CoV-GLUE [16], CoV-Spectrum [17] and Coronapp [18]. A further initiative based on the ENA dataset is the Galaxy project COVID-19 [19]; other systems use regional data: CLIMB-COVID (COG-UK) [20] and CovRadar [21]. The COVID-19 Data Portal [22] is provided by EMBL-EBI and the European COVID-19 Data Platform to facilitate data sharing and accelerate research through making all the data available in the public domain and encouraging the research community to share SARS-CoV-2 data. Furthermore, there are some open-source pipelines to identify mutations on SARS-CoV-2 data, i.e., Cecret [23], nf-core viralrecon [24], ncov2019-artic-nf [25],ViralFlow [26], Havoc [27], NCBI SARS-CoV-2 Variant Calling (SC2VC) [28] and ASPICoV [29].

Each dataset has its own advantages. While genomic assemblies are easier to share and interpret, raw reads provide granular information about the mutations through access to the pileup of reads supporting each mutation, also allowing the characterization of intrahost mutations. Analyzing both datasets together may support the identification of potential false positives in the data and the confirmation of trends.

To enable monitoring of SARS-CoV-2 sequences from both sources, we have developed CoVigator, an NGS pipeline and dashboard that allows geographical and temporal navigation through SARS-CoV-2 genomic variants. We automatically download and analyze genomic assemblies from the COVID-19 Data Portal and raw reads from the European Nucleotide Archive (ENA). Furthermore, we screen the literature for studies on SARS-CoV-2 intrahost mutations [30,31,32,33,34,35,36,37,38,39,40,41,42,43,44,45] and propose a filtering strategy to obtain a high-quality set of intrahost mutations in the large and heterogeneous dataset obtained from ENA. Thus, the CoVigator platform supports the early detection of variants that can potentially serve as the basis for further research in the field of vaccines.

## 2. Materials and Methods

The CoVigator knowledge base is implemented in Python version 3.8 and the database for storing data is PostgreSQL (version 13.4).

The CoVigator pipeline (version 0.14.0) is implemented in the Nextflow framework version 19.10.0. All dependencies are managed within conda (version 4.9) environments [46] (see Table 1). The pipeline may receive as input either (1) a single-end FASTQ, (2) two paired-end FASTQs, (3) an assembly in FASTA format or (4) a VCF file with mutations. Adapter sequences are trimmed from FASTQs using fastp [47], alignment to the reference genome is performed with BWA mem2 [48], Base Quality Score Recalibration (BQSR) is performed with GATK [49], duplicate reads are marked with sambamba [50] and, finally, a horizontal and vertical coverage analysis is performed with samtools [51]. Variant calling on the BAM files derived from the FASTQs is performed with LoFreq [52], GATK [49], BCFtools [53] and iVar [54] (only the results from LoFreq are shown in the dashboard). Variant calling on the FASTA assemblies is performed with a custom script using Biopython’s Needleman–Wunsch global alignment [55]. Further processing of VCF files adds functional annotations with SnpEff [56], technical annotations with VAFator [14], ConsHMM conservation scores [57] and Pfam protein domains [58]. Pangolin [59] is employed to determine the lineage of every sample. The input for pangolin is either the input assembly in FASTA format or the consensus assembly derived from the clonal mutations (i.e., VAF ≥ 0.8) and the reference genome. See Table 1 for more details on the specific settings of each tool.

The CoVigator dashboard is also implemented in Python using the visualization framework Dash (version 2.1.0). The computation is distributed through a high-performance computing cluster with a library that provides advanced parallelism, Dask (version 2022.9.2).

## 3. Results and Discussion

### 3.1. System Description

The CoVigator system (Figure 1) has three main components: (1) the knowledge base, (2) the analysis pipeline and (3) the dashboard. For every sample, the knowledge base orchestrates the metadata retrieval, raw data download and finally its analysis through the pipeline for the detection of mutations. Furthermore, it makes all necessary data available through a database (Postgre-SQL version 13). Finally, the dashboard presents the data to the end user through a set of interactive visualizations.

CoVigator operates via interaction with external systems: a high-performance computing (HPC) cluster and the ENA and COVID-19 Data Portal Application Programming Interfaces (APIs). Samples between both original datasets (raw reads and genomic assemblies) may overlap. As recommended, some data providers might automatically upload both data formats. The results presented in the dashboard are stratified by dataset.

### 3.2. Knowledge Base

The CoVigator knowledge base collects data from both genomic assemblies and raw reads, orchestrates its processing through the variant calling pipeline and stores all the metadata, raw data and processed results in a relational database.

The data for both datatypes are fetched via the corresponding API hosted by the European Bioinformatics Institute [60]; the metadata are normalized, the FASTQ (raw NGS reads) and FASTA (genomic assemblies) files are downloaded and their MD5 checksums are confirmed to ensure data integrity.

Furthermore, the knowledge base iteratively builds a variant co-occurrence matrix (only for the raw read dataset) and precomputes analyses of the data (binned abundance of mutations, dN/dS ratios per gene and domain, top occurring variants, pairwise co-occurrence and counts of variants per lineage, country, sample, mutation type, length and nucleotide substitution) that ensure low-latency responses.

### 3.3. Analysis Pipeline

In general, the CoVigator pipeline processes FASTQ and FASTA files into annotated and normalized analysis-ready VCF files via two independent workflows (Figure 1 and Appendix A). We implemented the pipeline in the Nextflow framework [61] and managed all dependencies with Conda environments to enable seamless installation. We have embedded the SARS-CoV-2 reference genome ASM985889v3 [5]. Using a different reference, this pipeline could instantly analyze other virus sequences as well.

### 3.4. Dashboard

The dashboard is the user interface to CoVigator. There are two separate views for the raw reads and genomic assembly datasets. Each view provides a set of tabs that allows the user to explore different aspects of the data held in the database. Each tab provides some interactive visualizations, described below. When applicable, the tabs provide a set of filters on the left side. These have been excluded from the screenshots for the purpose of clarity.

The most relevant tabs are described below, and some notable findings are highlighted. The data shown here include 137,025 samples downloaded from ENA on 21 October 2022 and 6,165,681 samples downloaded from the COVID-19 Data Portal on 18 November 2022.

### 3.5. Samples

The samples tab (Figure 2) enables the user to explore the accumulation of samples through time and the evolution of the dN/dS ratio in different genomic regions.

Figure 2A shows the accumulation of samples in each country. The dashboard allows the user to select specific countries and/or lineages.

Figure 2B,C show the dynamics of the dN/dS ratio through time, over genes and protein domains. The dN/dS ratio aims to estimate the evolutionary pressure on SARS-CoV-2 proteins and domains. This metric, although originally developed for assessing diverging species, is an imperfect but simple estimation of the evolutionary pressure within the same species [62,63], in this case, SARS-CoV-2. There have been recent efforts to develop better alternatives for estimating the evolutionary pressure on SARS-CoV-2 [64]. The traditional interpretation of dN/dS is as follows: dN/dS < 0 indicates purifying selection, dN/dS = 1 indicates neutral evolution and dN/dS > 1 indicates positive selection.

### 3.6. Lineages

The lineages tab enables the user to explore the different lineages through time and geography (Figure 3). Both the accumulation of samples in every lineage worldwide (Figure 3A) and the dominant lineage through time (Figure 3B) can be viewed. In the screenshot, the displacement of B.1 by Alpha (B.1.1.7), the subsequently displacement by multiple Delta lineages (AY.*) and finally displacement by the three Omicron lineages (BA.1, BA.2 and BA.3) can be seen.

### 3.7. Mutation Statistics

The mutation statistics tab provides insights into the variant calling results on the different datasets and genomic regions (Figure 4). Expected trends in the data can be confirmed in these visualizations.

The median number of SNVs per sample in the raw read dataset is 32, with an interquartile range (IQR) of 30 (Figure 4A). Additionally, the median number of MNVs is two with an IQR of one. The number of deletions is lower (median: 3, IQR: 2) than the number of SNVs, and the number of insertions is even lower, with few samples having just one insertion. For the genomic assemblies, the numbers are slightly different, with median SNVs = 44 (IQR: 19), MNVs = 2 (IQR: 0), deletions = 4 (IQR: 3) and again just one insertion in a few samples (Figure 4B).

We observe that the base substitution C > T is by and large the most frequent, followed by G > T and A > G; the deletion TA > T and the MNV GG > AA is the most frequent in both datasets (Figure 4C,D).

In Figure 4E,F, we confirm that deletions are more frequent than insertions with an insertion-to-deletion ratio of 0.002 and 0.032 for raw reads and genomic assemblies, respectively. We also confirm two previous findings: (1) shorter deletions and insertions are more common than longer ones [65,66] and (2) the deletions and insertions not causing a frameshift are overrepresented as their impact in the resulting protein are more subtle [67]. In the genomic assemblies, we observe a long tail of deletions longer than 8 bp, which is not observed in the raw read results. We suspect this is a technical artefact introduced via our variant calling method. Finally, as shown in Figure 4G,H we observe that the most frequent mutation effect is a missense variant, followed by a synonymous variant. This is coherent between both datasets.

### 3.8. Recurrent Mutations

The recurrent mutations tab allows the user to explore the most recurrent mutations by the total count of observations through time within their genomic context (Figure 5). In Figure 5A, the top recurrent mutations and their frequency and counts through time are shown. The size of the table can be parametrized for up to 100 mutations. For instance, the user can explore the most recurrent mutations in the whole genome, a given gene or a given protein domain. Furthermore, the period in which the monthly counts are shown can be parameterized. The gene viewer (Figure 5B) has multiple tracks: (i) a scatter plot with the relevant mutations and their frequencies in the virus population, (ii) ConsHMM conservation tracks and (iii) gene and Pfam protein domains. The table in Appendix A provides the decline and rise of the Alpha and Delta lineages, respectively, in the counts of mutations between April and July 2021.

Additionally, the mutation statistics tab provides a co-occurrence analysis that points to clusters of co-occurring mutations and their correspondence with virus lineages; or in the case of mutations shared between lineages, these clusters may contain a mixture of different but related lineages. Due to performance limitations, this analysis is only available in the raw read dataset and at the gene level. In Appendix A, we show the Jaccard index co-occurrence matrix in the spike protein and its clustering results annotated with SARS-CoV-2 lineages in Appendix A.

### 3.9. Clonal and Intrahost Mutations in the Raw Read Dataset

The FASTQ files provide the pile-up of reads across the genome, and this gives detailed information into the called variants. In particular, we can count the number of reads supporting each variant, and this allows us to identify subclonal variants supported using only a fraction of the reads. These variants likely emerged within the host and are referred to as intrahost variants. The identification of intrahost variants is not possible on the genomic assemblies.

We consider high-quality clonal mutations as those with a VAF greater than or equal to 80%, and those with a VAF greater than or equal to 50% and lower than 80% as low-confidence clonal mutations. Only high-confidence clonal mutations are used to determine a consensus sequence and assign a SARS-CoV-2 lineage (Figure 6).

The remaining dataset of mutations poses a different technical challenge due to the difficulty of separating true low VAF mutations from noise. We first determine those mutations with a VAF below 50% as raw candidate intrahost mutations.

We observed a large number of low-frequency mutations among SARS-CoV-2 genomes. In order to establish a high-quality set of intrahost mutations for studying viral evolution, we screened and compared the literature on SARS-CoV-2 intrahost mutations for different filtering approaches and implemented a conservative approach (Table 2).

## 4. Conclusions

The persistently increasing amount of publicly available SARS-CoV-2 sequencing data calls for robust platforms that allow constant monitoring of genomic SARS-CoV-2 variants in heterogeneous data sets. Our CoVigator pipeline covers the essential steps of preparing the data and calling variants from SARS-CoV-2 raw sequencing data from ENA and genome assemblies from the COVID-19 Data Portal. The pipeline is integrated within the CoVigator knowledge base that orchestrates the download, processing and storage of the underlying samples and results. The CoVigator dashboard provides different visualizations and features for selecting clonal variants across all genes from the SARS-CoV-2 genome in a selected period. The dashboard also provides a comprehensive analysis of intrahost variants observed across detected mutations in the raw read dataset. To this end, we propose a conservative filtering approach based on filtering samples and mutations. The dataset of intrahost mutations derived from public data that we make available through CoVigator is, to the best of our knowledge, the largest published dataset of SARS-CoV-2 intrahost mutations. The main strength of CoVigator is the combination of a software pipeline with a dashboard, which ensures both processing of the data and its interpretation. Uniquely, CoVigator processes genome assemblies and raw sequencing data types, making it open-data-friendly and allowing it to be adopted to other SARS-CoV-2 data sources. A brief comparison of the important features of CoVigator with other pipelines is tabulated in Appendix A.

The identification of mutations over such heterogeneous datasets obtained with different sequencing protocols is challenging. With CoVigator, we observed VAF dilution on mutations identified via targeted amplicon sequencing with overlapping primers, genome edge effects and read edge effects. We aim to address these challenges in the future, e.g., through inferring the primers used in an arbitrary sample. Additionally, we implemented a simplistic phasing of clonal mutations occurring in the same amino acid to ensure their correct annotation. However, we identified the need for a phasing method for low-VAF mutations that existing germline phasing tools do not cover. CoVigator is currently limited to processing Illumina sequencing data, while the majority of SARS-CoV-2 sequencing projects (i.e., Artic network) and pipelines use Oxford Nanopore sequencing. SARS-CoV-2 Nanopore data processing will be implemented in subsequent releases of CoVigator.

Future versions of CoVigator can be broadened to other use cases, such as other infectious organisms or co-existing infections during the pandemic (see Appendix A for further details [69,70]). Additionally, we envision the annotation of all possible mutations before their observation to potentially improve preparation for future variants of concern.

## Figures and Tables

**Figure 1 viruses-15-01391-f001:**
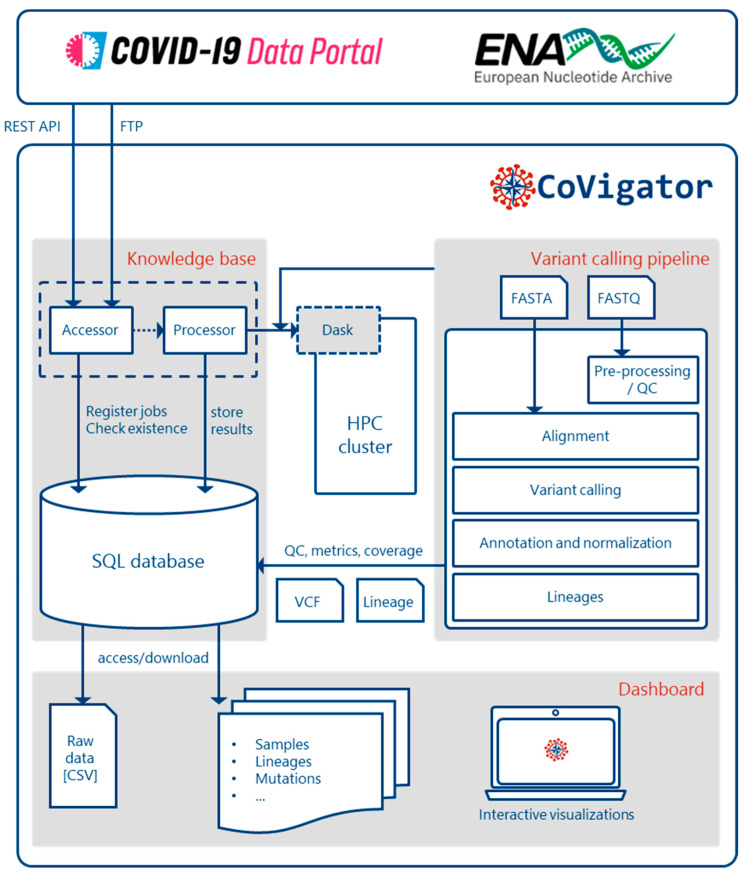
CoVigator system components. The accessor reads external data and stores it in an SQL database. The processor reads the stored data and distributes the processing of every sample in an HPC cluster via Dask. The pipeline processes FASTA and FASTQ data and finally stores the results back in the database (See Appendix A for a more detailed FASTA and FASTQ processing pipeline). The dashboard reads the results and displays them in a set of interactive plots. The results are also available in raw format.

**Figure 2 viruses-15-01391-f002:**
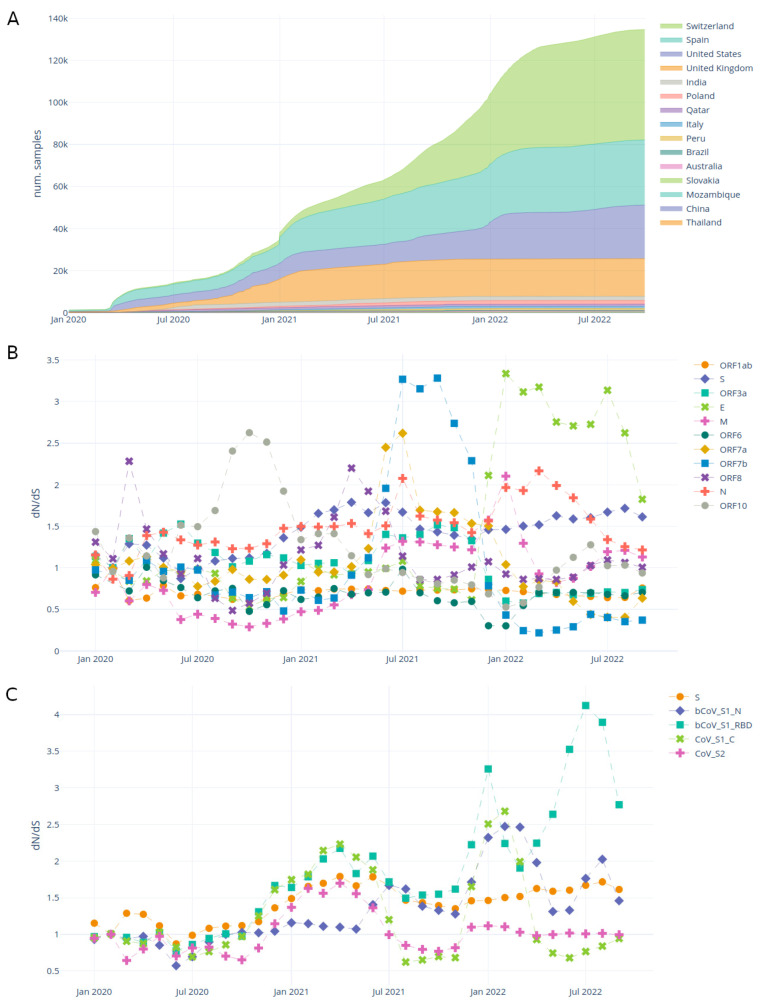
Samples by country tab plots for raw read dataset. (**A**) accumulation of samples through time by country; (**B**) dN/dS ratio through time on each SARS-CoV-2 protein; (**C**) dN/dS ratio through time in the domains of the spike protein. See Appendix A for a screenshot including the filters.

**Figure 3 viruses-15-01391-f003:**
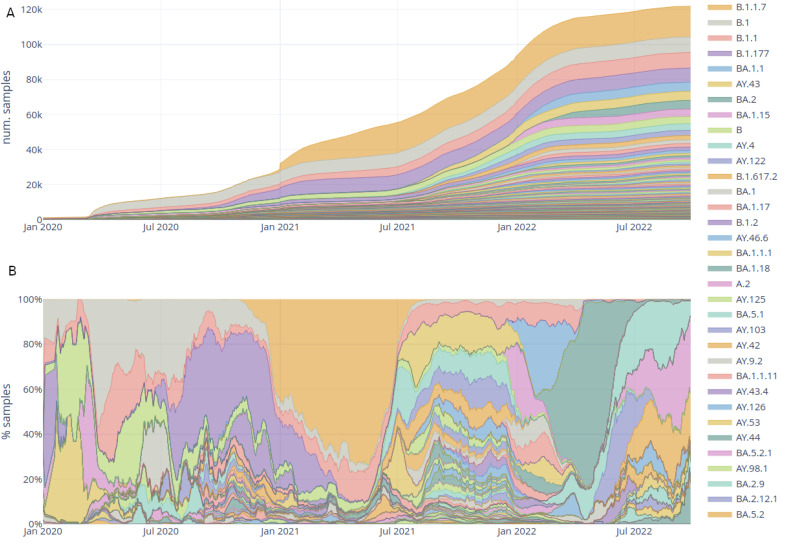
Interactive plots in the lineages tab for the raw read dataset. (**A**) Accumulation of samples in each lineage through time; (**B**) dominant lineages through time. See Appendix A for a screenshot including the filters.

**Figure 4 viruses-15-01391-f004:**
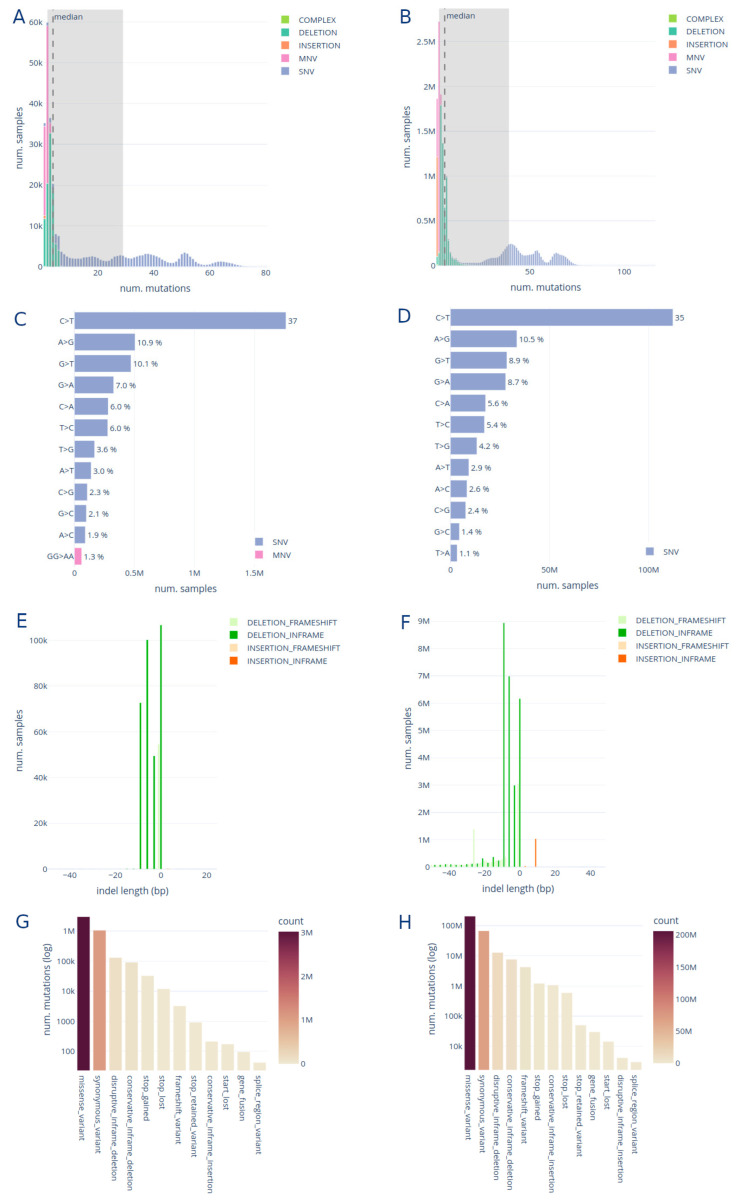
Interactive plots on the mutation statistics tab showing results for raw reads and genomic assembly datasets. (**A**) ENA distribution of the number of mutations per sample; (**B**) C19DP distribution of the number of mutations per sample; (**C**) ENA frequency of base substitutions, (**D**) C19DP frequency of base substitutions; (**E**) ENA indel length distribution; (**F**) C19DP indel length distribution; (**G**) ENA frequency of mutation effect on the protein; (**H**) C19DP frequency of mutation effect on the protein. See Appendix A for a screenshot including the filters.

**Figure 5 viruses-15-01391-f005:**
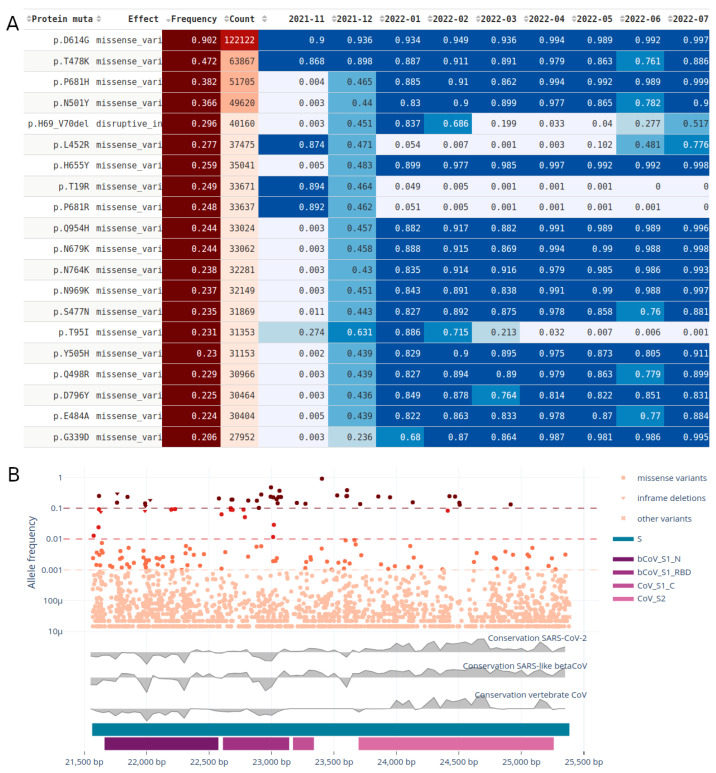
Gene view for the spike protein on the raw read dataset. (**A**) Table of the top 20 recurrent mutations with the frequency segregated by month between November 2021 and July 2022; (**B**) gene view showing mutations (synonymous and unique mutations excluded) in the spike protein and their frequencies in the virus population, the ConsHMM conservation tracks in grey and the Pfam protein domains in tones of purple. See Appendix A for a screenshot including the filters.

**Figure 6 viruses-15-01391-f006:**
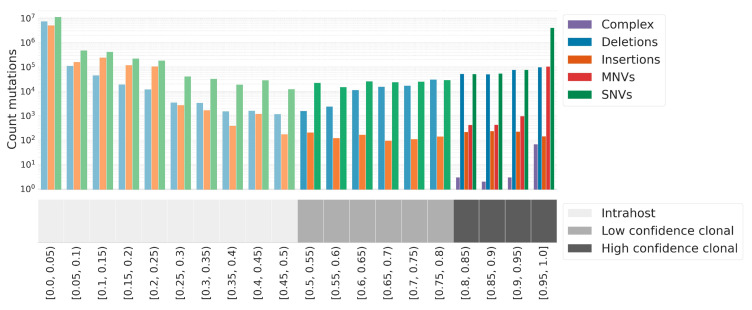
Distribution of VAF across all mutation calls (4,665,192 with VAF ≥ 0.8; 222,297 with VAF ≥ 0.5 and < 0.8; 26,231,409 with VAF < 0.5) in 135,347 samples. High-confidence clonal mutations overlapping the same amino acid are merged into MNVs or complex variants. See Appendix A for a screenshot of intrahost mutations tab including the filters.

**Table 1 viruses-15-01391-t001:** Tools employed in the pipeline, specific versions and settings.

Tool	Purpose	Settings	References	Version	FASTQ	FASTA
fastp	Adapter trimming		[47]	0.20.1	X	
BWA mem 2	Alignment	Default	[48]	2.2.1	X	
GATK	Variant calling and alignments preprocessing	MQ ≥ 20, BQ ≥ 20, ploidy = 1	[49]	4.2.0.0	X	
sambamba	Read deduplication	MQ ≥ 20, BQ ≥ 20, ploidy = 1	[50]	0.8.2	X	
samtools	Coverage analysis		[51]	1.12	X	
LoFreq	Variant calling	MQ ≥ 20, BQ ≥ 20	[52]	2.1.5	X	
BCFtools	Variant calling, normalization and annotation	MQ ≥ 20, BQ ≥ 20	[53]	1.14	X	X
iVar	Variant calling	MQ ≥ 20, BQ ≥ 20	[54]	1.3.1	X	
Biopython	Custom variant calling on assemblies sequences based on Needleman-Wunsch global alignment	aligner.mode = ‘global’ aligner.match = 2 aligner.mismatch = −1 aligner.open_gap_score = −3 aligner.extend_gap_score = −0.1 aligner.target_end_gap_score = 0.0 aligner.query_end_gap_score = 0.0	[55]	1.79		X
SnpEff	Functional annotations		[56]	5.0	X	X
VAFator	Technical annotations	MQ > 0, BQ > 0	[14]	1.2.5	X	
Pangolin	Lineage calling		[59]	4.1.2	X	X
ConsHMM	Conservation annotations		[57]	Not available	X	X
Pfam	SARS-CoV-2 protein domains		[58]	Not available	X	X

**Table 2 viruses-15-01391-t002:** Published and implemented filtering approaches for intrahost variants.

Approach	Sample Filters	Variant Filters
Valesano-like [44]	≥50,000 mapped reads ≥29,000 bp horizontal coverage	VAF ≥ 2%, VAF < 50% DP ≥ 100 ≥10 supporting reads
Sapoval-like [39]	≥20,000 mapped reads	VAF ≥ 2%, VAF < 50% DP ≥ 10 Mask extremes of genome + homoplasmic positions [68]
Tonkin-Hill-like [38]	Excessive number iSNVs (99.9th percentile) Outlier number of iSNVs with mid-VAFs, between 40% and 80%	VAF ≥ 5%, VAF < 50% DP ≥ 100 ≥5 supporting reads
CoVigator approach	≥50,000 mapped reads ≥29,000 bp horizontal coverage Excessive number iSNVs (99.9th percentile) Outlier number of iSNVs with mid-VAFs, between 40% and 80%	VAF ≥ 2%, VAF < 50% DP ≥ 100 ≥10 supporting reads Mask extremes of genome + homoplasmic positions [68] from indels ≤ 10 bp

## Data Availability

The CoVigator dashboard is accessible via covigator.tron-mainz.de and can be installed via https://github.com/TRON-bioinformatics/covigator. A standalone version of the CoVigator pipeline with nextflow is available at https://github.com/TRON-Bioinformatics/covigator-ngs-pipeline. CoVigator documentation is available at https://covigator.readthedocs.io. All links accessed on 16 June 2023.

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
