# Peer review of "CoVigator—A Knowledge Base for Navigating SARS-CoV-2 Genomic Variants"

_viruses, 2023, doi:10.3390/v15061391_

Round 1
Reviewer 1 Report
This is a well written and organized description of a new database tool to explore SARS-Cov-2 variants.
Introduction and methods are clear and concise. It would greatly improve the manuscript if the CoVigator pipeline is presented in greater detail as supplementary information or in an extended version of the methods (current methods version is too brief and not very informative).
It would also improve the manuscript by incorporating in the discussion a few lines highlighting the differences and strengths of CoVigator compared to other efforts already published to create accesible databases on coronavirus.
In general a very good manuscript.
Reviewer 2 Report
The authors have developed a resource of currently identified mutations in the SARS-CoV-2 genome and offer tools how one might track newly arising mutations of the virus. They imply that in this way the scientific community could be alerted in a timely frame to newly arising potentially dangerous mutations. The latter endeavor appears rather expression of hope than a reality. However, the overall value of this carefully updated collection of data on new mutations and variants cannot be overestimated. The methods recommended by the authors to further expand their data collection have been realistically chosen.
In their conclusions the authors might want to consider tuning down their enthusiasm.
The frequently used term "pipeline" sounds strange in a scientific report. Could you replace that word for readers mainly interested in the science part of publications. Thank you.
Reviewer 3 Report
Manuscript titled "CoVigator - a knowledge base for navigating SARS-CoV-2 genomic variants" (viruses-2451791)is a great piece of work. Overall the draft is is in good shape and I do not find any major issue that prevent acceptance of this work. However I think author just improve following minor things before draft is officially accepted for publication.
1 Minor writing issue
2 Quality of figure need to improve, difficult to read information within figures.
3 Author should discuss implication of this work for other infectious diseases means need to develop similar approach for other diseases
4 Challenges and weakness of the developed tool is missing need to mention those
Writing need minor work
